# Lymphoid B cells upregulate HIV-1 *ex vivo* and are linked to its expression *in vivo*

Matthew T. Ollerton[1]*, Joy M. Folkvord[1], Veronica Bush[2], David A. Parry[3], Amie L. Meditz[4], Martin D. McCarter[5], Fred Yost[6], Cecilia M. Shikuma[6], Elizabeth Connick[1]

1 Department of Medicine, Division of Infectious Diseases, The University of Arizona College of Medicine, Tucson, Arizona, United States of America, 2 Department of Physiology and Medical Sciences, The University of Arizona College of Medicine Tucson, Tucson, Arizona, United States of America, 3 Department of Otolaryngology, The University of Arizona College of Medicine Tucson, Tucson, Arizona, United States of America, 4 Beacon Center for Infectious Diseases, Boulder, Colorado, United States of America, 5 Department of Surgery, University of Colorado Anschutz Medical Campus, Aurora, Colorado, United States of America, 6 Hawaii Center for HIV/AIDS, University of Hawai'i at Manoa John A Burns School of Medicine, Honolulu, Hawaii, United States of America

* ollerton.matthew@mayo.edu

## Abstract

B cell follicles in secondary lymphoid tissues are major sites of HIV expression in untreated people living with HIV (PLWH), but whether lymphoid B cells promote HIV expression in CD4+T cells is unknown. Using a tonsil model and flow cytometry, germinal center B cells induced HIV expression in follicular helper CD4+T cells (TFH) in a concentration-, contact-, and major histocompatibility complex class II (MHC-II)-dependent manner. Non-naïve tonsil B cells also induced HIV expression in nonTFH CD4+T cells that was MHC-II restricted. *In situ* hybridization and immunofluorescent staining of lymph node sections from six PLWH on long-term antiretroviral therapy (ART) and six ART-naïve PLWH demonstrated most HIV RNA-expressing (vRNA+) cells were adjacent to at least one B cell. In both groups, vRNA+ cells per mm² were elevated in B cell follicles, particularly after adjusting for CD3+CD4+ frequencies. TFH (PD-1+*BCL6*+) were a minority of vRNA+ cells in all PLWH. In contrast to ART-naïve PLWH, most vRNA+ cells in PLWH on ART resided outside follicles, and were preferentially adjacent to extrafollicular B cells. Thus, B cells upregulate HIV expression largely through cognate interactions *in vitro* and are linked to HIV expression in secondary lymphoid tissues.

## Author summary

HIV expression is concentrated primarily in secondary lymphoid tissues in people living with HIV both on and off antiretroviral therapy. This study demonstrates B cells induce HIV predominately through cognate interactions. In tissues, most HIV-expressing cells are adjacent to B cells *in vivo,* even in regions with relatively fewer B cells.

**Data availability statement:** All data supporting the findings of this study are available within the article and its supplementary information files.

**Funding:** This work was supported in part by the National Institutes of Health, National Institute of Allergy and Infectious Diseases to E.C (R56AI181705), the Moya-Teller Fund at the University of Arizona to M.T.O, the Edwin C Cadman Endowment Fund at the University of Hawai'i to C.M.S. The funders had no role in study design, data collection and analysis, decision to publish, or preparation of the manuscript.

**Competing interests:** The authors have declared that no competing interests exist.

## Introduction

Most human immunodeficiency virus type 1 (HIV) and simian immunodeficiency virus (SIV) replication occurs in secondary lymphoid tissues (SLT) including lymph nodes, spleen, and mucosal associated lymphoid tissues such as those in the gut and tonsils [1]. CD4+ T cells located in B cell follicles and germinal centers (GC) are major sites of HIV and SIV replication during chronic, untreated infection prior to AIDS [2–5]. This has been attributed to multiple factors including the presence of infectious virions bound to follicular dendritic cells (FDC) in immune complexes [6], the highly permissive state of follicular T helper cells (TFH) [7], and the relative paucity of virus-specific CD8+ T cells in follicles and GC compared to extrafollicular regions [8–9]. After initiation of antiretroviral therapy (ART), FDC-bound SIV and HIV immune complexes, as well as virus-expressing CD4+ T cells, diminish over time [10]. Importantly, however, low frequencies of virus-expressing cells can be detected in SLT even after prolonged ART, despite undetectable plasma viral loads [1]. It is postulated that these cells are the main driver of viral rebound when ART is stopped and pose a major barrier to HIV cure [11–12]. Factors that induce and sustain HIV and SIV expression in SLT, especially during effective ART, and the phenotype of virus-expressing cells are not well understood.

B cell follicles are critical to induction of humoral immune responses and the site of residence for the majority of B cells in SLT [13]. B cells assist in the formation of GC and participate in antigen transport into follicles and onto FDC [14–15]. GC B cells (GCB) interact with TFH and FDC to promote the development of antibody responses to antigens, such as HIV, deposited on FDC [16]. Both FDC and TFH initiate the recruitment of CXCR5-expressing pre-TFH to follicles through secretion of the chemokine CXCL13 [17–18], and antigen presenting GCB recruit cognate pre-TFH from T:B borders into GC [19–22]. Antigen specific interactions with GCB advance TFH differentiation further by promoting expression of the transcription factor BCL6 and additional surface markers including PD-1 [22–24]. Whether GCB induce HIV expression in TFH, however, has never been investigated.

In the present study we first examined whether GCB or FDC enhance HIV expression in TFH using an *ex vivo* tonsil model of HIV infection that employs HIV GFP reporter viruses [7,25] in which GFP correlates with HIV production [26]. GCB, but not FDC, consistently induced HIV expression in TFH in a contact-dependent and dose-dependent manner. GCB also upregulated multiple pathways linked to antigen-specific stimulation in CD4+ T cells and their impact on HIV expression was restricted by major histocompatibility complex class II (MHC-II) blocking antibody. Remarkably, upregulation of HIV expression was not limited to GCB or TFH, as other non-naïve SLT B cell populations augmented HIV expression in both TFH and nonTFH CD4+ T cells, and their activity was also restricted by MHC-II blockade. The frequency, distribution and phenotype of HIV RNA-expressing (vRNA+) cells *in vivo* were determined in inguinal lymph node sections from people living with HIV (PLWH) on long-term ART and without ART. In PLWH on prolonged ART, most vRNA+ cells were located outside of B cell follicles, whereas vRNA+ cells predominated in B cell

follicles in PLWH not on ART. TFH, defined as PD-1[+]*BCL6*[+] cells, constituted a minority of vRNA[+] cells in PLWH, and significantly fewer vRNA[+] TFH were detected in those on ART than those not on ART. Heightened concentrations of vRNA[+] cells were found in B cell follicles in both groups of PLWH, particularly after normalization to frequencies of CD4[+] T cells in follicular and extrafollicular compartments. Most vRNA[+] cells were adjacent to at least one B cell in PLWH regardless of ART, and preferentially located adjacent to B cells in extrafollicular regions of lymph nodes in PLWH on ART. Collectively, these findings demonstrate that SLT B cells upregulate HIV expression predominately through cognate interactions and likely promote HIV expression *in vivo*.

## Results

### GCB augment HIV expression in TFH through direct contact *ex vivo*

Disaggregated human tonsil cells were infected by spinoculation with HIV GFP reporter viruses [7,25] and utilized to determine whether GC-resident cells, i.e., GCB and FDC, induce HIV expression in TFH. Sorted TFH (CD3[+]CD8[-]CXCR5[hi]PD-1[hi]) were infected with a replication competent X4-tropic HIV GFP reporter virus (X4-HIV) [27] and cultured alone or with FDC (CD45[-]CD35[+]), GCB (CD19[+]CD3[-]IgD[-]CD38[mid]), or both for three days in the presence of saquinavir to prevent spreading infection. Percentages of GFP[+] TFH as well as GFP MFI were significantly elevated in the presence of GCB compared to TFH cultured alone, whereas FDC alone or in combination with GCB did not significantly alter GFP expression (Fig 1A). Similar increases in percentages of GFP[+] TFH and GFP MFI induced by GCB were observed when an R5-tropic HIV GFP reporter virus (R5-HIV) was used (Fig 1B). In unsorted tonsil cells, B cell depletion reduced both percentages of GFP[+]CD3[+]CD8[-] cells and GFP MFI compared to undepleted tonsil cells when infected with either X4-HIV (Fig 1C) or R5-HIV (Fig 1D). Incubation of TFH with increasing numbers of GCB revealed that increases in both percentages and GFP MFI of GFP[+] cells were GCB dose-dependent (Fig 1E and S1A Fig) and all subsequent experiments utilized GCB:TFH at a 1:1 ratio. GFP expression in TFH was consistently elevated only when GCB were cultured directly with TFH, and not when separated by a semi-permeable membrane (Fig 1F). Total and integrated HIV DNA levels in TFH cultures did not differ significantly in the presence or absence of GCB after 18 hours (Fig 1G). Whereas the number of viable uninfected labeled TFH was higher than GCB after three days in culture with HIV-spinoculated TFH (S1B Fig), viable spinoculated TFH cell numbers were not significantly altered by the presence of GCB (Fig 1H), weighing against enhanced proliferation or viability as mechanisms underlying GCB upregulation of HIV expression in TFH.

### GCB, but not HIV, alter expression of multiple genes in TFH

To determine the effect of GCB and HIV on TFH gene expression, TFH isolated from six tonsils were spinoculated with either DMEM media (mock) or R5-HIV and cultured in the presence of saquinavir and uninfected labeled TFH (TFH + TFH) or GCB (TFH + GCB). After three days, live TFH were purified by FACS and RNA was isolated and analyzed for gene expression using the 785 gene nCounter Host Response Panel. As before, GCB augmented HIV expression in TFH in these experiments (S2A Fig). Using the Benjamini Hochberg adjusted p-value and a 1.5-fold change threshold, no genes were significantly altered in either the R5-HIV spinoculated TFH + TFH cultures compared to the mock-spinoculated TFH + TFH cultures (S2B Fig), or the R5-HIV spinoculated TFH + GCB cultures compared to mock-spinoculated TFH + GCB cultures (S2C Fig), indicating that HIV infection did not significantly alter expression of the genes evaluated. In contrast, 60 genes were significantly altered ≥1.5-fold in the mock-spinoculated TFH + GCB cultures compared to mock-spincoculated TFH + TFH cultures (Fig 2A) and 38 genes in the R5-HIV spinoculated TFH + GCB cultures compared to the R5-HIV spinoculated TFH + TFH cultures (Fig 2B), revealing a widespread effect of GCB on TFH gene expression, regardless of HIV infection. In a similar experiment utilizing six separate tonsils and TFH spinoculated with X4-HIV, GCB significantly altered 116 genes (S2D Fig). Of the genes significantly altered by GCB in R5-HIV infected TFH, 84% were shared by the mock-spinoculated cultures and 92% were shared by X4-HIV spinoculated cultures (S2E and S2F Fig), indicating similar effects of GCB on TFH in the context of X4-HIV infections. Among the genes most significantly

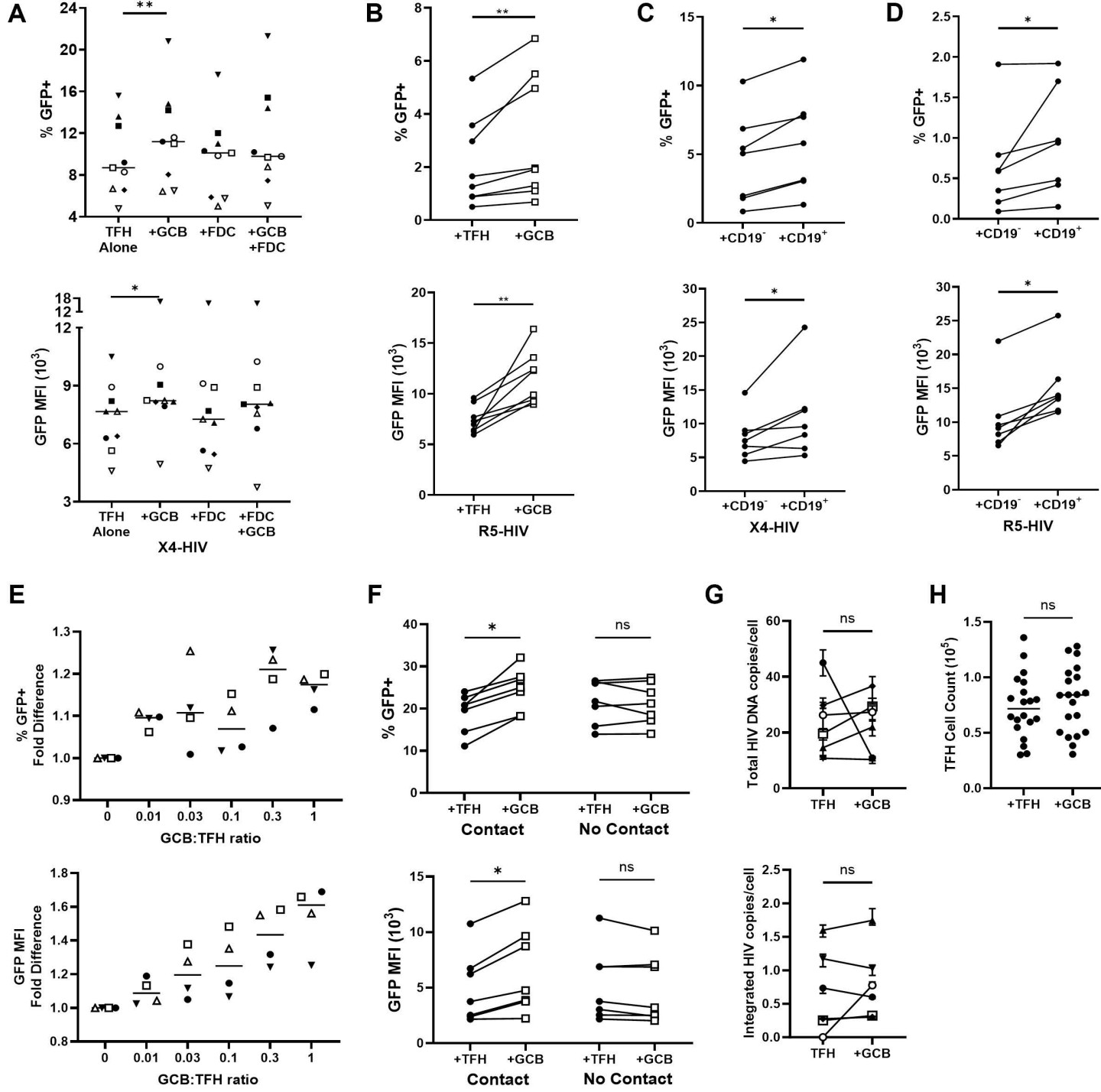

**Fig 1. Tonsil GCB augment HIV replication in TFH *ex vivo*.** (A) TFH (CD3⁺CD8⁻CD19⁻CXCR5^hiPD-1^hi) were spinoculated with X4-HIV GFP reporter virus (X4-HIV) and cultured with autologous GCB (CD19⁺CD3⁻CD38^midIgD⁻) and FDC (CD19⁻CD45⁻CD35⁺) for 3 days in R-15 and 5µM saquinavir. Percent GFP⁺TFH and GFP MFI of GFP⁺TFH were determined by flow cytometry (n = 9). (B) TFH were spinoculated with R5-HIV GFP reporter virus (R5-HIV) and 1x10⁶ spinoculated cells were cultured at a ratio of 1:1 with either autologous uninfected, CellTrace Blue labeled TFH or GCB for 3 days. Graphs represent percent GFP⁺TFH in CellTrace⁻CD3⁺ cells and GFP MFI of GFP⁺TFH (n = 8). (C-D) CD19⁺ cells were depleted from disaggregated tonsil cells using CD19 magnetic beads. 5x10⁶ CD19⁻ cells were spinoculated with X4-HIV (C) or R5-HIV (D) and cultured 3 days with autologous uninfected, dye-labeled CD19⁻ or CD19⁺ cells at sample-specific ratios of CD19⁻:CD19⁺ cells. Graphs depict percent GFP⁺CellTrace⁻CD3⁺CD8⁻ cells and

GFP MFI of GFP$^+$CellTrace$^-$CD3$^+$CD8$^-$ cells (n = 7). (E) 2x10$^5$ TFH were spinoculated with X4-HIV and cultured with autologous uninfected CellTrace Blue labeled GCB at the ratios indicated. Percent GFP$^+$ TFH and GFP MFI of GFP$^+$ TFH were determined by flow cytometry and reported as fold differences relative to TFH alone (n = 4). (F) 1x10$^6$ X4-HIV spinoculated TFH were cultured with autologous uninfected, dye-labeled TFH or GCB at a ratio of 1:1 for 3 days either in direct contact (+contact) or separated by a 0.4μm transwell insert (no contact). Graphs depict percent GFP$^+$CellTrace$^-$TFH and GFP MFI of GFP$^+$CellTrace$^-$TFH (n = 7). (G) 2x10$^6$ TFH were spinoculated with X4-HIV and cultured in the absence or presence of autologous GCB at a 1:1 ratio for 18 hours. GCB and TFH cultured alone were combined immediately prior to DNA isolation to ensure equivalent cell numbers. Cellular DNA was used to determine total and integrated HIV DNA copies per cell (n = 6). (H) X4-HIV spinoculated TFH were cultured with autologous uninfected, dye-labeled TFH or GCB for 3 days and total numbers of CellTrace$^-$TFH were determined using absolute counting beads by flow cytometry (n = 20). Horizontal bars indicate medians (A,E). Error bars indicate ranges (G). Friedman tests and Dunn's multiple comparison tests were used for (A) and Wilcoxon tests were used for (B-D, F-I) as determined by Graphpad Prism v10 with significance indicated: ns, not significant; *p < 0.05; **p < 0.01.

and consistently upregulated were those involved in T-cell costimulatory pathways including *TNFRSF9* encoding 4–1BB, *TNFRSF4* encoding OX40, and *TNFRSF18* encoding GITR, all of which can be induced by TCR signaling. Furthermore, several signaling pathways downstream of TCR signaling including T cell costimulation, IL-2, NF-kappa B (NFκB), and JAK-STAT were significantly upregulated in the presence of GCB (Fig 2C).

## MHC-II blocking antibody reduces GCB-induced HIV expression in TFH

The tonsil model was employed to examine potential mechanisms for upregulation of HIV replication in TFH by GCB suggested by the gene expression studies. To exclude the possibility that the CD3 antibody used for TFH isolation facilitated GCB augmentation of HIV expression in TFH, TFH were isolated with or without CD3 antibody prior to spinoculation and culture. No impact on GCB-induced HIV expression in TFH was detected when CD3 antibody was omitted (S2G Fig). ICAM-1, ICOS, and CD40L are important for TFH interactions with B cells in mice [20,28,29] and humans [30–31]; however, attempts to manipulate interactions between GCB and TFH through blocking antibodies to ICAM-1 and CD40, and soluble ICOS and CD40L did not abolish GCB-mediated upregulation of HIV expression in TFH (S2H and S2I Fig). Despite evidence that GCB induce expression of *IL2, TNFRSF9,* and *TNFRSF18* noted above, blockade of these pathways with IL-2 neutralizing antibody, 4–1BB-Fc, or anti-GITRL did not reduce GCB mediated effects on HIV expression (S2J Fig). Since T cell costimulation with 4–1BB [32], GITR [33], or OX40 [34] induces noncanonical NFκB signaling, and noncanonical NFκB activation through AZD5582 has been proposed as a latency reversal agent in vivo [35–36], we evaluated whether AZD5582 and the noncanonical NFκB inhibitor NIK-SMI1 alter GCB-mediated upregulation of HIV in TFH. Despite altered HIV expression in TFH by AZD5582 and NIK-SMI1, no significant differences in GCB-mediated effects on HIV expression were observed (S2K Fig). The JAK inhibitors ruxolitinib and tofacitinib inhibit HIV expression in PBMC derived CD4$^+$T cells [37]; however, GCB mediated upregulation of HIV expression in TFH occurred in the presence and absence of JAK inhibitors (S2L Fig). Blocking cognate interactions using the MHC-II antibody IVA12, on the other hand, significantly impaired GCB-mediated induction of HIV expression in TFH (Fig 2D and S2M Fig). Taken together, these data suggested that several pathways downstream of TCR signaling did not impact GCB-induced HIV expression in TFH when evaluated individually. But cognate interactions between TFH and GCB were important for GCB induced HIV expression.

## HIV and GCB downregulate CXCR5 expression on TFH

We previously reported that the follicular homing molecule, CXCR5, is downregulated on X4-HIV spinoculated CD3$^+$CD8$^-$ cells from tonsils [7]. We evaluated here whether HIV and GCB affect cell surface CXCR5 expression on HIV-expressing TFH *ex vivo*. When X4-HIV spinoculated TFH were cultured with saquinavir in the presence of labeled, uninfected TFH or GCB for three days, CXCR5 expression was significantly lower on GFP$^+$ TFH compared to GFP$^{neg}$ TFH in both cultures, extending our prior observations (Fig 3A). CXCR5 receptor internalization due to binding of its ligand CXCL13 is one potential mechanism of reduced surface CXCR5 expression [38], and it is notable that GCB induced CXCL13 expression in the aforementioned gene expression experiments (Fig 2A, 2B and S2D Fig). CXCL13 concentrations were quantified

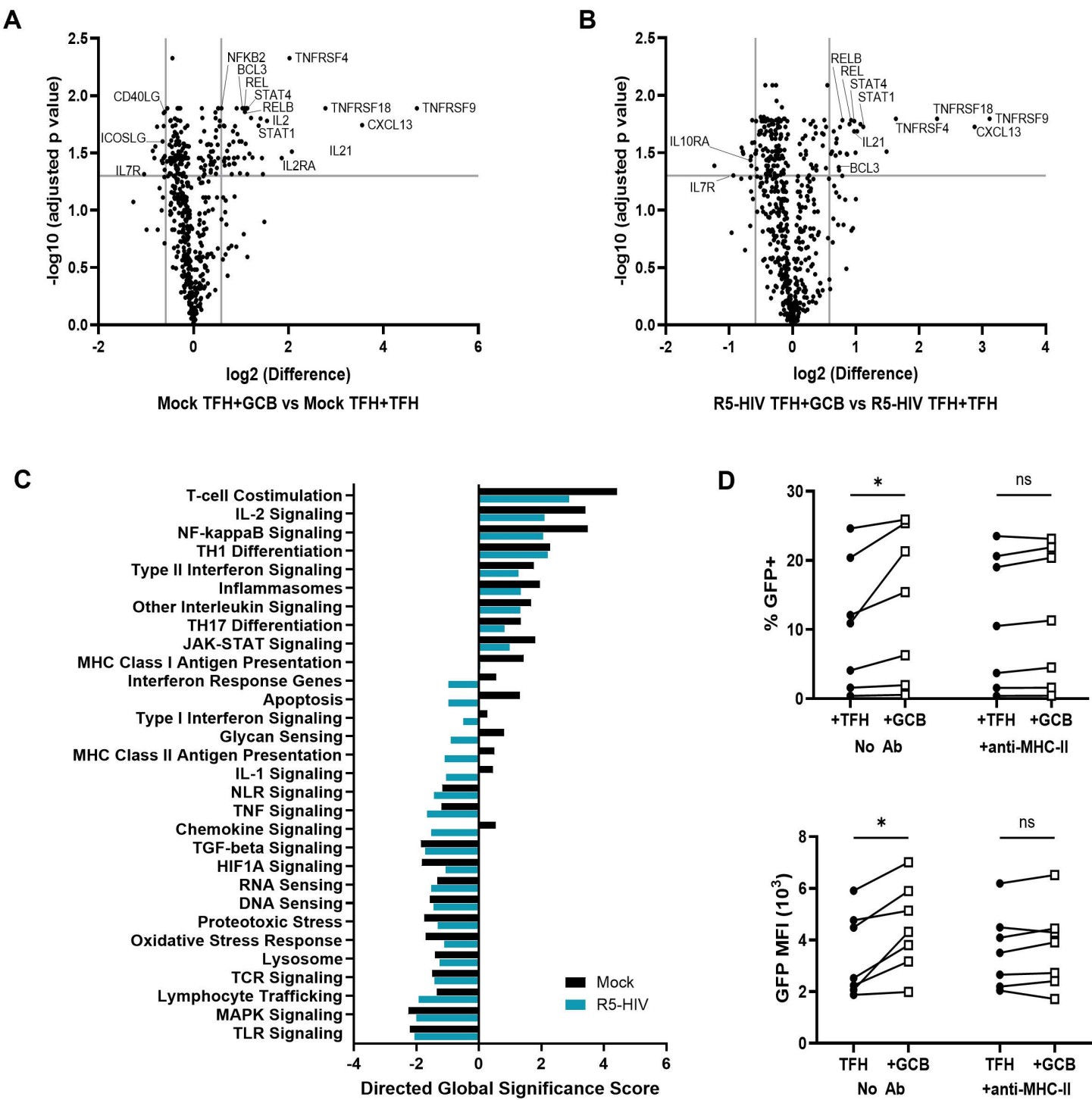

**Fig 2. Tonsil GCB alter TFH gene expression and MHC-II blockade reduces GCB-induced HIV expression in TFH.** (A-B) TFH (CD3+CD8-CD19-CXCR5hiPD-1hi) from six tonsils were spinoculated with either media (mock) or R5-HIV GFP reporter virus (R5-HIV). 4x10^6 spinoculated TFH were cultured with 4x10^6 autologous uninfected, violet proliferation dye labeled TFH or GCB (CD19+CD3-CD38midIgD-) for 3 days in R-15 with 5μM saquinavir. Live 7-AAD-violet proliferation dye-TFH were isolated by FACS. Total RNA was isolated and gene expression analysis was performed using Nanostring's host response panel on the nCounter. Volcano plots depict differentially expressed genes in TFH from mock (A) and R5-HIV (B) cultures that were cultured with GCB compared to TFH cultured with uninfected, labeled TFH. Vertical lines represent 1.5 fold change and horizontal lines represent an

adjusted p value of 0.05, as determined using the Benjamini Hochberg method. (C) Directed global significance scores were determined for the 30 pathways in which ≥15 genes from the host response panel were assigned using nSolver 4.0 advanced analysis. (D) TFH were spinoculated with X4-HIV GFP reporter virus and cultured with autologous uninfected, dye-labeled TFH or GCB for three days in the absence or presence of anti-MHC-II antibody. Percent GFP+ TFH (top) and GFP MFI of GFP+ TFH (bottom) were determined by flow cytometry (n = 7). Statistical analyses were determined using Wilcoxon matched paired test as determined by Graphpad Prism v10 and significance indicated: ns, not significant; *p < 0.05.

from supernatants of X4-HIV spinoculated TFH cultured for three days with either uninfected, labeled TFH or GCB (Fig 3B). CXCL13 supernatant expression was normalized to live TFH cell numbers determined using viability dye and counting beads in flow cytometric analyses of the cultured cells. GCB induced CXCL13 expression 1.8-fold compared to

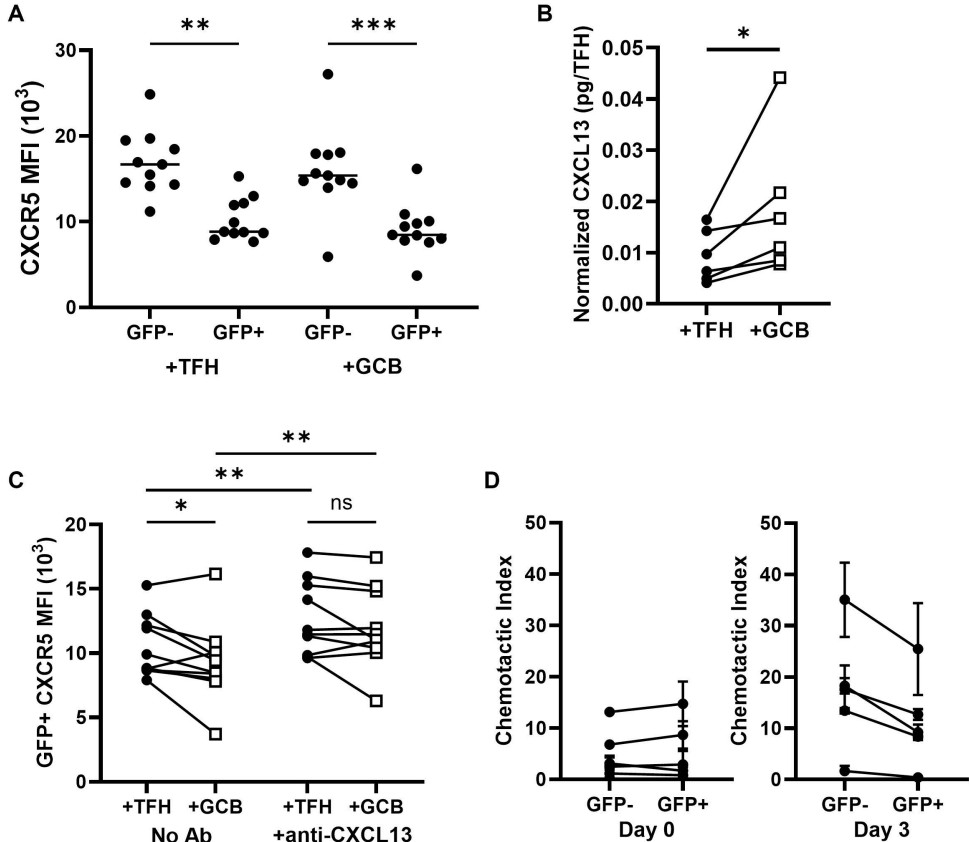

**Fig 3. HIV and tonsil GCB downregulate CXCR5 on TFH.** (A) TFH (CD3+CD8-CD19-CXCR5hiPD-1hi) were spinoculated with X4-HIV GFP reporter virus (X4-HIV) and cultured with autologous uninfected, CellTrace Blue labeled TFH or GCB (CD19+CD3-CD38midIgD-) for 3 days in R-15 and 5μM saquinavir. CXCR5 MFI of CellTrace-GFP+ and GFP- TFH was determined by flow cytometry (n = 11). (B) Supernatant from cultures of TFH spinoculated with X4-HIV and cultured with autologous uninfected CellTrace Blue labeled TFH or GCB were collected after three days. Live, CellTrace Blue-TFH cell counts were determined by flow cytometry. CXCL13 concentrations were determined by ELISA and normalized to pg/TFH (n = 6). (C) CXCR5 MFI was determined by flow cytometry on GFP+CellTrace-TFH from cultures of X4-HIV spinoculated TFH cultured with autologous uninfected, CellTrace Blue labeled TFH or GCB in the presence or absence of CXCL13 blocking antibody after 3 days (n = 10). (D) TFH (CD3+CD8-PD-1hiICOShi) were spinoculated with X4-HIV, then recovered for 1 hour in R-15 with 5μM saquinavir. Half of the cells were cultured 3 days in R-15 with 5μM saquinavir, while the other half were washed and subjected to a CXCL13 chemotaxis assay using 5 μm transwells. After 4 hours, cells were collected from both top and bottom wells, washed, and cultured 3 days in R-15 containing 5μM saquinavir. On day 3, the chemotaxis assay was performed on the remaining TFH untouched post spinoculation. GFP expression was determined in TFH from top and bottom wells and total GFP+ and GFP- cells counts were determined using absolute count beads by flow cytometry to determine chemotactic indexes (n = 5). Horizontal bars indicate medians (A). Wilcoxon matched paired tests were performed using Graphpad Prism v10 and significance indicated: ns, not significant; *p < 0.05; **p < 0.01; ***p < 0.001.

cultures without GCB (Fig 3B). Blockade of CXCL13 with a neutralizing antibody elevated CXCR5 MFI in GFP+ TFH, and abrogated GCB-mediated downregulation of CXCR5 on GFP+ TFH (Fig 3C), demonstrating that CXCR5 downregulation by GCB was induced by CXCL13.

To determine whether HIV expression reduced TFH chemotaxis to CXCL13, TFH were sorted from tonsils using PD-1 and ICOS co-expression, to avoid potentially confounding effects of CXCR5 antibody (S3 Fig). Chemotaxis assays were performed on days 0 or 3 after infection with X4-HIV and cells were cultured in the presence of saquinavir to prevent spreading infection. GFP expression and chemotactic indexes were calculated based on flow cytometry data collected on day 3 for all experiments. When chemotaxis assays were performed on day 0, chemotactic indexes were similar between GFP+ and GFP- cells (Fig 3D). When chemotaxis assays were performed on day 3, chemotactic indexes were reduced in GFP+- compared to GFP TFH, suggesting that HIV expression reduced TFH migration to CXCL13 (Fig 3D).

## Multiple SLT B cell subsets upregulate HIV expression in CD4+ T cells *ex vivo*

We next evaluated whether upregulation of HIV expression is unique to GCB and TFH interactions or if it occurs when other subsets of SLT B and CD4+ T cells are cultured together. TFH and nonTFH CD4+ T cells were spinoculated with X4-HIV and cultured in media supplemented with saquinavir and either uninfected labeled TFH or nonTFH, respectively, or one of four B cell subsets sorted based on IgD and CD38 expression. Representative flow plots of gating strategies used for sorting CD4+ T cell and B cell subsets are shown in Fig 4A. The following subsets were predominately represented by sorted B cell populations: GCB (IgD-CD38$^{mid}$), pre-GCB (IgD+CD38+), memory B cells (IgD-CD38-), and naïve B cells (IgD+CD38-) [39]. GFP expression was evaluated after 3 days. Percentages and MFI of GFP were higher for TFH (Fig 4B) compared to nonTFH (Fig 4C), consistent with the well-established heightened permissivity of TFH [2,7,40]. Most B cell subsets augmented GFP expression in TFH and nonTFH. The primary exception was the IgD+CD38- naïve subset, which did not significantly impact GFP expression in TFH and only increased GFP percentages in nonTFH. MHC-II blocking antibody significantly impaired B cell-induced GFP expression in nonTFH (Fig 4D). Gene expression analysis of R5-infected nonTFH cultured with nonGCB identified 15 genes significantly altered by nonGCB cells (S4A Fig). Similar to the effect of GCB on TFH, nonGCB induced T cell costimulation, IL-2 signaling, and NFκB signaling in nonTFH (S4B Fig), and all but two nonTFH genes altered significantly by nonGCB were shared with TFH + GCB cultures (S4C Fig). Collectively, these data demonstrate that multiple SLT B cell subsets, particularly non-naïve B cells, upregulate HIV expression in TFH and nonTFH CD4+ T cells predominately through cognate interactions.

## The frequency, distribution, and phenotype of SLT vRNA+ cells differ between PLWH based on ART status

Earlier studies demonstrated that B cell follicles in lymph nodes of untreated, chronically infected PLWH prior to AIDS harbor the majority of vRNA+ cells and that TFH preferentially express HIV RNA [7,9]. Whether B cell follicles harbor the majority of vRNA+ cells and the role of TFH in vRNA expression in PLWH on long-term ART has not been studied *in situ*. Inguinal lymph nodes from six males LWH on prolonged ART with virologic suppression and six ART-naïve males LWH were evaluated by in situ hybridization and immunostaining to determine the frequency and phenotype of vRNA+ cells and their distribution within lymph node tissue. Groups were well matched in terms of demographic and clinical characteristics except for age (Table 1).

Several differences in tissue morphology and vRNA distribution observed between PLWH on ART and ART-naive PLWH were anticipated based on prior studies. Tissue sections were smaller from PLWH on ART than those not on ART (Fig 5A), consistent with observations that lymphadenopathy usually resolves in the context of ART [41]. Percentages of tissue area that consisted of follicle tended to be lower (Fig 5B), and percentages of B cell follicles that consisted of GC were significantly lower in PLWH on ART compared to those not on ART (Fig 5C), in accordance with known decreases in follicular hyperplasia in the context of ART [42]. Frequencies of vRNA+ cells were significantly lower in PLWH on ART compared to PLWH not on ART (Fig 5D), reflecting the antiviral effects of ART. VRNA+ particles were readily detected in

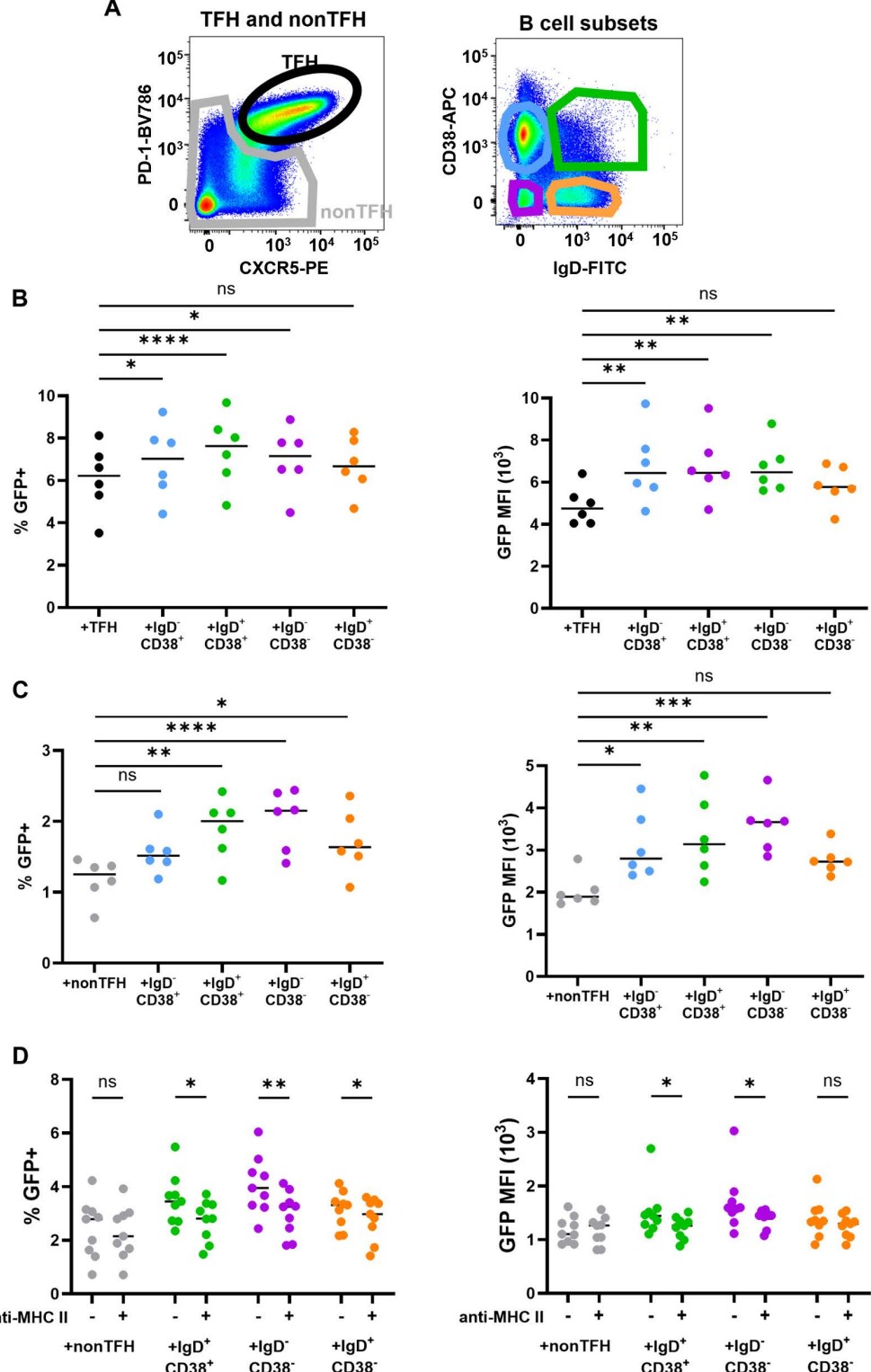

**Fig 4. Multiple tonsil B cell subsets upregulate HIV expression in CD4+ T cells *ex vivo*.** (A) Gating strategies for TFH (CD3+CD8-CD19-CXCR5hiPD-1hi) and nonTFH (CD3+CD8-CD19-CXCR5+/-PD-1+/-) (left) and autologous B cell populations from single, live CD3-CD19+ cells using IgD and CD38 (right) are shown in representative flow cytometry plots. (B) TFH were spinoculated with X4-HIV GFP reporter virus and cultured with autologous uninfected,

CellTrace Blue labeled TFH or B cell subsets at a ratio of 1:1 for 3 days in R-15 with 5µM saquinavir. Percent GFP+CellTrace-T cells and GFP MFI of GFP+CellTrace-T cells were determined by flow cytometry. (C) NonTFH were cultured with CellTrace Blue labeled nonTFH or B cell subsets and analyzed, as described in (B). (D) NonTFH were spinoculated with X4-HIV GFP reporter virus and cultured with autologous uninfected, dye-labeled nonTFH or B cell subsets for three days in the absence or presence of anti-MHC-II antibody. Percent GFP+CellTrace-nonTFH and GFP MFI of GFP+CellTrace-nonTFH were determined by flow cytometry (n = 9). Horizontal bars indicate medians (B-D).Statistical analyses were determined using Friedman's test with uncorrected Dunn's multiple comparisons (C-D) or Wilcoxon match paired tests using Graphpad Prism v10 and significance indicated: *p < 0.05; **p ≤ 0.01; ***p ≤ 0.001; ****p ≤ 0.0001.

**Table 1. Demographics and clinical characteristics of male inguinal lymph node donors.**

| Subject ID | Age[a] | Race | Viral Load (log$_{10}$ copies/mL)[b] | CD4 (Cells/mm³)[c] | Years on ART | ART Regimen[d] |
|---|---|---|---|---|---|---|
| **PLWH: No ART** | | | | | | |
| LN23 | 32 | Caucasian | 4.00 | 726 | – | – |
| LN25 | 34 | Caucasian | 4.36 | 553 | – | – |
| LN78 | 25 | Caucasian | 3.82 | 1103 | – | – |
| LN106 | 40 | African American | 5.61 | 706 | – | – |
| LN132 | 45 | Hispanic | 5.79 | 479 | – | – |
| LN136 | 31 | African American | 5.24 | 300 | – | – |
| **PLWH: +ART** | | | | | | |
| LN-01 | 48 | African American | Undetectable | 1039 | 25 | EVG/c, FTC, TDF |
| LN-04 | 63 | Caucasian | Undetectable | 1153 | 18 | DRV/r, TDF |
| LN-05 | 59 | Asian | Undetectable | 462 | 7 | EVG/c, FTC, TAF |
| LN-07 | 54 | Caucasian | Undetectable | 860 | 29 | RAL, FTC, TAF |
| LN-08 | 61 | African American | Undetectable | 223 | 20 | EFV, FTC, TDF |
| LN-09 | 66 | Caucasian | Undetectable | 538 | 21 | RAL, FTC, TAF |

[a]No ART vs. +ART: p=0.0022.

[b]Limit of assay detection in PLWH +ART: 20 copies/mL.

[c]No ART vs. +ART: p=0.94.

[d]EVG/c, elvitegravir boosted with cobicistat; FTC, emtricitabine; TDF, tenofovir disoproxil fumarate; DRV/r, darunavir boosted with ritonavir; RAL, raltegravir; TAF, tenofovir alafenamide; EFV, efavirenz.

association with FDC in PLWH not on ART, as previously described[7,43–44], but no vRNA+ particles were detected in association with FDC in lymph node sections of PLWH on long-term ART, consistent with prior findings that vRNA particles on FDC decay in the context of ART [10].

Multiple differences in tissue distribution and phenotype of vRNA+ cells were observed between PLWH on and not on ART that were not anticipated. A minority (median, 41%) of vRNA+ cells were located in B cell follicles in PLWH on ART, whereas the majority (median, 73%) were located in B cell follicles in PLWH not on ART (Fig 5E). Percentages of vRNA+ TFH, determined by colocalization of vRNA, *BCL6* and PD-1, as shown in representative images (S5A Fig), were a minority in both groups, and significantly lower in PLWH on ART (median, 5%) compared to PLWH not on ART (median 27%; Fig 5F). PD-1+vRNA+ cells also constituted a minority of vRNA+ cells in PLWH on ART (median, 8%) and those not on ART (median, 41%) (S5B Fig). Comparisons between percentages of CD3+*CD4*+ cells that were TFH (S5C Fig) and percentages of vRNA+ cells that were TFH within follicles and extrafollicular regions, revealed that TFH preferentially expressed vRNA only in B cell follicles of PLWH not on ART (Fig 5G). There was no evidence of preferential vRNA expression in TFH in follicles of PLWH on ART or in extrafollicular compartments in either group.

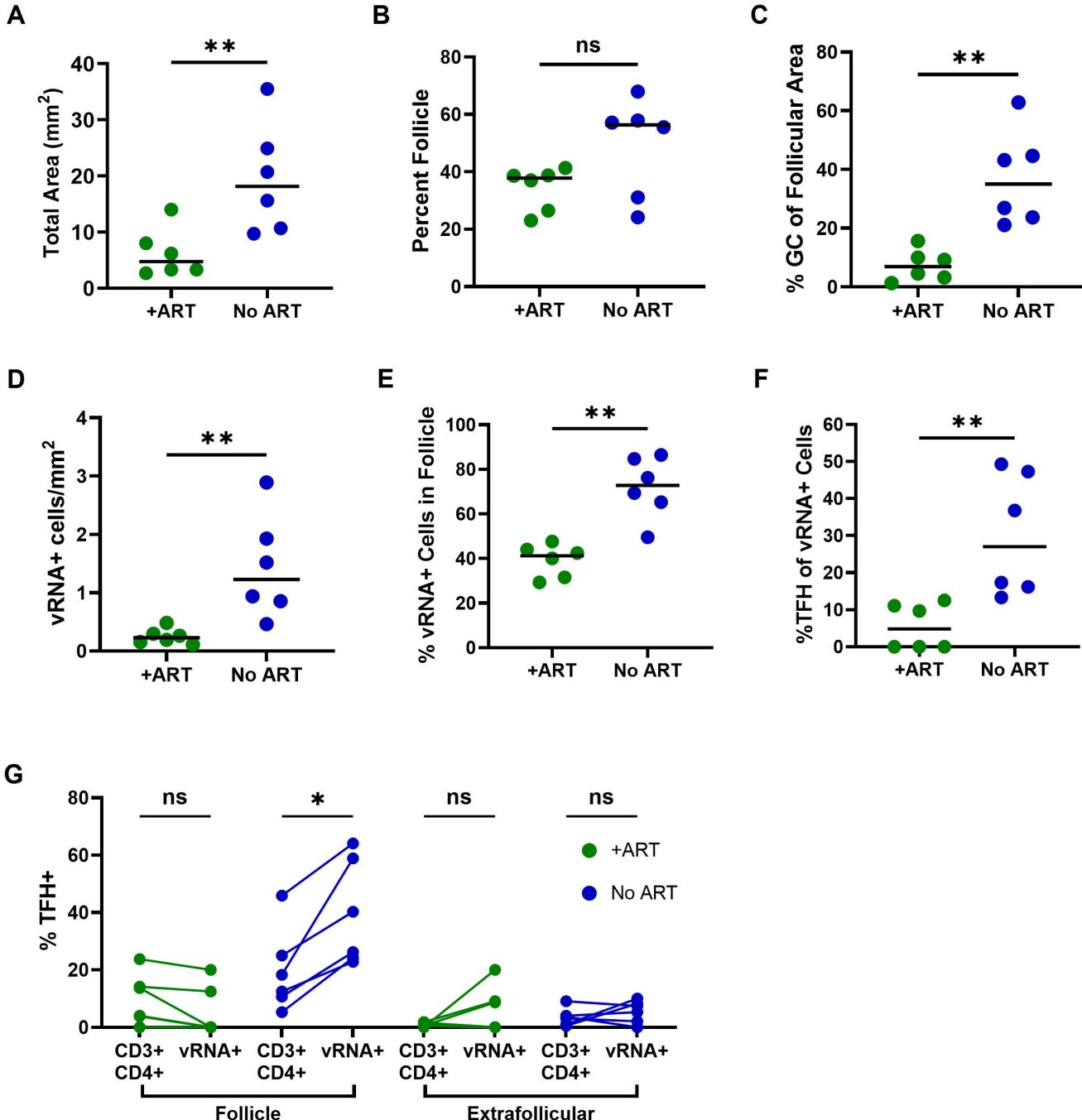

**Fig 5. The frequency, distribution and phenotype of SLT vRNA⁺ cells differ between PLWH based on ART status.** Lymph node sections from 6 PLWH on ART (+ART) and 6 ART-naïve PLWH (no ART) were evaluated for vRNA, *BCL6*, and *CD4* by *in situ* hybridization, CD3, PD-1, FDC and CD20 by immunofluorescence staining, and nuclei by DAPI staining using visual inspection and quantitative image analysis. (A) Total DAPI⁺ cross sectional tissue area. (B) Percent tissue that is follicle, as determined by CD20 staining. (C) Percent of follicle area that consisted of GC, as determined by FDC staining. (D) Frequencies of vRNA⁺DAPI⁺ cells per mm² of tissue. (E) Percentages of vRNA⁺DAPI⁺ cells located within follicles. (F) Percentages of vRNA⁺ cells that were vRNA⁺ TFH (vRNA⁺*BCL6*⁺PD-1⁺). (G) Percentages of CD3⁺CD4⁺ cells that were TFH (CD3⁺PD-1⁺) compared to percentages of vRNA⁺ cells that were TFH (vRNA⁺*BCL6*⁺PD-1⁺) within follicles and extrafollicular regions of PLWH. Horizontal bars indicate medians (A-F). Numbers of cells evaluated to determine percentages (E-G) are reported in S1 Table. Statistical analyses of unpaired samples (A-F) were performed using Mann-Whitney tests and of paired samples (G) were determined using Wilcoxon tests as determined by Graphpad Prism v10 and significance indicated: ns, not significant; *p < 0.05; **p < 0.01.

PLOS Pathogens

**B cells are preferentially located adjacent to vRNA⁺ cells in lymph nodes of PLWH**

The tonsil model revealed that SLT B cells upregulate HIV expression in CD4⁺ T cells and that this is both dose- and contact-dependent. We next evaluated whether physical adjacency between SLT B cells and vRNA⁺ cells supports the hypothesis that B cells promote HIV expression *in vivo*. Each vRNA⁺ cell was evaluated visually for direct contact with B cells, as shown in representative images (Fig 6A-6D). Frequencies of CD20⁺ B cells, as determined by CD20 immunostaining and quantitative image analysis were not significantly different in follicular regions of PLWH on and not on ART (Fig 6E), although B cell frequencies were significantly lower in extrafollicular regions of PLWH on ART compared to those not on ART (Fig 6F). Almost all vRNA⁺ cells in B cell follicles were located adjacent to multiple CD20⁺ cells, as shown in representative images (Fig 6A and 6B). Most vRNA⁺ cells in PLWH on ART (median, 67%) and not on ART (median, 78%) were in direct contact with B cells (Fig 6G). Frequencies of vRNA⁺ cells were significantly higher in B cell follicles compared to extrafollicular regions in all PLWH (Fig 6H). After adjusting for frequencies of CD3⁺*CD4⁺* cells in each compartment (S5D Fig), the magnitude of differences between follicular and extrafollicular concentrations of vRNA⁺ cells was even more pronounced in both groups (Fig 6G). We next evaluated extrafollicular regions to determine whether vRNA⁺ cells were preferentially located adjacent to B cells. The probability of a vRNA⁺ cell being adjacent to at least one B cell was calculated assuming a random distribution of B cells within the extrafollicular tissue. In PLWH on ART, vRNA⁺ cells were consistently more likely to be found adjacent to B cells than estimated by random B cell distribution in extrafollicular regions, whereas no such predisposition was seen in PLWH not on ART (Fig 6H). These findings demonstrate that adjacency to B cells is linked to HIV expression in follicular regions of SLT in all PLWH, and in extrafollicular regions of lymph nodes in PLWH on ART.

## Discussion

This is the first study to demonstrate that SLT B cells upregulate HIV expression in CD4⁺ T cells. GCB consistently augmented HIV expression in TFH using an *ex vivo* model of HIV infection in SLT that has been previously shown to recapitulate multiple aspects of *in vivo* infection [7,25]. The effects of GCB were both dose- and contact-dependent. GCB upregulated multiple pathways linked to HIV expression, and the effects were due to GCB and not to HIV infection. Furthermore, upregulation of HIV expression was dependent on MHC-II, and not on noncanonical NFκB signaling, 4–1BB, CD40L, GITR, or ICAM-1. Importantly, upregulation of HIV expression was not limited to GCB and TFH. Non-naïve B cell subsets within SLT upregulated HIV expression in both TFH and nonTFH, and this was also MHC-II dependent. SLT from PLWH demonstrated marked differences in the distribution and phenotype of vRNA⁺ cells between those on prolonged ART and those ART-naive. Most vRNA⁺ cells in PLWH not on ART were located within B cells follicles, whereas the majority of vRNA⁺ cells were located in extrafollicular regions in PLWH on ART. TFH constituted a minority of vRNA⁺ cells in both groups, but were significantly lower in those on ART. Most vRNA⁺ cells in both groups of PLWH were located adjacent to at least one B cell and B cell follicles were associated with heightened frequencies of vRNA⁺ cells, particularly after adjusting for frequencies of CD4⁺ T cells. Remarkably, vRNA⁺ cells were more frequently found adjacent to B cells than expected by chance in extrafollicular regions of all PLWH on ART, providing circumstantial evidence that B cells are linked to HIV expression in extrafollicular regions during ART as well. Collectively, these data suggest that B cells play a significant role in inducing HIV RNA expression throughout SLT both in the presence and absence of ART, and this is likely mediated by antigen-specific interactions.

While it has been known for 36 years that antigen presentation signaling through T cell receptors (TCR) activates HIV expression in CD4⁺ T cells [45], this is the first report showing that SLT B cells augment HIV expression and do so predominately through TCR and MHC-II interactions. The classical professional antigen presenting cells, dendritic cells and macrophages, on the other hand, have been shown to induce HIV expression *in vitro* in prior studies [46–48]. The extent to which antigen presentation by any of these cells induces HIV expression *in vivo* is unknown. Several lines of evidence,

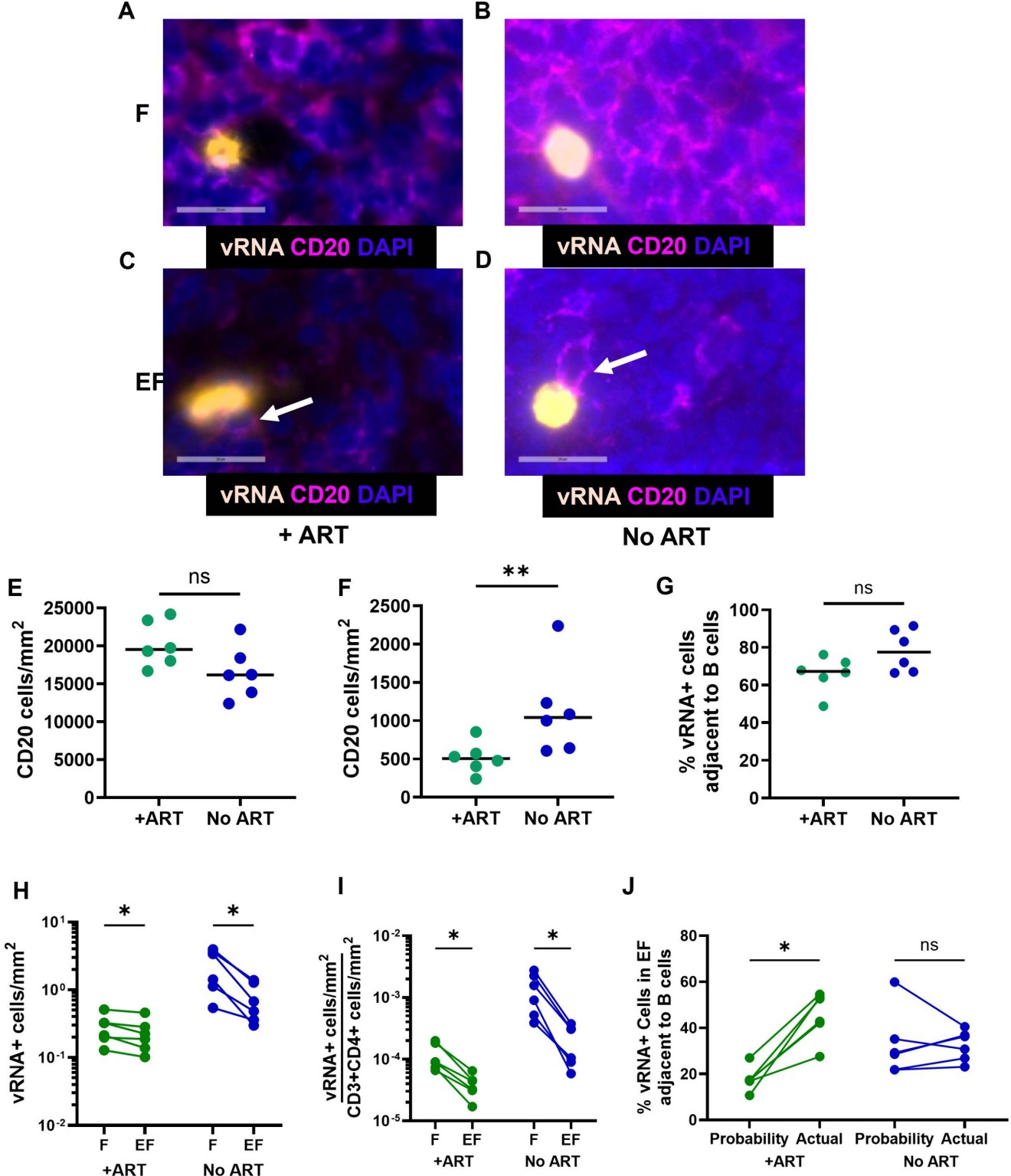

**Fig 6. B cells are linked to vRNA⁺ cells *in vivo* in PLWH.** Lymph node sections from 6 PLWH on ART (+ART) and 6 ART-naïve PLWH (no ART) were evaluated for vRNA and CD4 by *in situ* hybridization, CD3 and CD20 by immunofluorescence staining, and nuclei by DAPI staining and analyzed by visual inspection and quantitative image analysis. (A-D) Representative images of vRNA⁺(gold) DAPI⁺(blue) cells in follicles (F) (A, B) or extrafollicular

regions (EF) (C, D) adjacent to CD20+ (magenta) DAPI+ cell in PLWH on ART (A, C) or not on ART (B, D). A single plane of the z-stacked vRNA image is shown. (E, F) Frequencies of CD20+DAPI+ cells in follicular (E) and extrafollicular (F) regions. (G) Percentages of total vRNA+DAPI+ cells directly adjacent to at least one CD20+B cell. (H) vRNA+DAPI+ cell frequencies in follicular (F) and extrafollicular (EF) areas defined by CD20 staining. (I) vRNA+ cell frequencies were normalized to CD3+CD4+frequencies in follicular and extrafollicular areas. (J) The probability of a vRNA+ cell being adjacent to at least one CD20+DAPI+ cell within extrafollicular regions was determined using the frequency of CD20+DAPI+ cells in extrafollicular regions as reported in (6F) and assuming a random distribution (Probability). Percentages of extrafollicular vRNA+DAPI+ cells adjacent to at least one CD20+DAPI+ cell as determined by visual inspection were reported (Actual). Horizontal bars indicate medians (E-G). Mann-Whitney tests were performed for unpaired comparisons (E-G), and Wilcoxon tests were used for paired comparisons (H-J) as determined by Graphpad Prism v10 and significance indicated: ns, not significant; *p < 0.05; **p < 0.01.

however, suggest that antigen presentation by B cells is likely a major stimulus for HIV expression in PLWH. *Ex vivo*, we demonstrated that SLT B cells upregulated both HIV expression and key T cell gene transcripts associated with antigen stimulation including OX40, 4–1BB, CD69, and CD25 [49–50]. Although other B cell factors may promote HIV expression, blockade of several known receptor ligand pairs between antigen-stimulated TFH and GCB did not impair GCB-induced HIV expression when evaluated individually, suggesting that HIV expression occurred concomitantly with antigen presentation, not in response to ancillary pathways induced by GCB including costimulatory and cytokine signaling pathways. Importantly, these markers and pathways may represent a signature of B cell–induced TCR activation within HIV-expressing cells *in vivo.* Furthermore, the finding that naïve B cells were less adept at promoting HIV expression in SLT CD4+T cells is consistent with antigen-specific stimulation playing a major role in induction of HIV expression. Elevated levels of HIV expression in B cell follicles in PLWH regardless of ART status are also highly suggestive of the importance of antigen-specific stimulation since follicular CD4+T cells gain access to B cell follicles through cognate interactions with B cells [51–52] and TFH engage B cells through TCR-MHC-II interactions in GC [53]. Although a previous study suggested that FDC enhance HIV expression in CD4+T cells through secretion of TNF-alpha [54], FDC did not induce HIV expression in TFH in our experiments. Furthermore, prior transcriptomic analyses showed that *TNF* is not expressed by FDC at appreciable levels [16,55]. Thus, within B cell follicles, antigen-specific stimulation by B cells is likely a major driver of HIV expression.

Whether vRNA+ cells in extrafollicular regions are induced by cognate interactions is not clear. In extrafollicular regions of PLWH on prolonged ART, vRNA+ cells were significantly more likely to be located adjacent to a B cell than expected based on the frequencies of B cells in those regions. These studies likely underestimate the true frequency of vRNA+ cells in contact with B cells because adjacency was only determined in two dimensions and, furthermore, lymphocytes within SLT are mobile. The *in situ* data presented here, however, do not define whether vRNA+ cells were stimulated by antigen presentation. Identification of a B cell-induced signature of antigen presentation in vRNA+ cells could be useful in further cementing this potential relationship. In PLWH not on ART, there was no evidence of a relationship between B cell adjacency and extrafollicular vRNA expression. It is possible that cognate interactions are less important in the induction of virus expression in the context of the highly inflammatory environment and high levels of extracellular virus present in SLT in the absence of ART. Alternatively, in individuals not on ART, macrophages and dendritic cells may play a more important role in upregulating HIV expression in extrafollicular regions than B cells. Future studies to evaluate the relationship between macrophages and dendritic cells in extrafollicular tissues *vis a vis* vRNA+ cells could potentially provide insight into this question.

Surprisingly, a minority of vRNA+ cells were TFH in both groups of PLWH, in contrast to prior reports that TFH are the major vRNA+ cells in PLWH on [42] and not on ART [2]. Most prior studies have assumed that vRNA+ cells were TFH based on measurements of vRNA in sorted cell populations [2,42] or on their location in follicles or GC [3,7,9,56,57]. This is the first study to identify HIV RNA+TFH *in situ* by co-expression of PD-1 and the canonical transcription factor Bcl6. Previous studies in HIV seronegative tonsils demonstrated diverse CD4+T cell phenotypes exist, even within specialized GC [58–59], and that various memory T cells contribute to GC responses [60]. Thus, location alone is insufficient to identify

TFH. Interestingly, in the present study, significantly lower percentages of vRNA+ cells were TFH in individuals on long-term ART compared to those not on ART. Banga *et al.*[42] reported elevated levels of cell-associated (CA) vRNA as well as high amounts of replication competent HIV as determined by a viral outgrowth assay in PD-1+ lymph node cells from PLWH on ART [42], but these levels declined after eight years of ART. The duration of ART ranged from 0.3 to 14 years in the study by Banga *et al.,* whereas participants in the present study had received long-term ART ranging from seven to 29 years. Collectively, these data suggest that a prolonged duration of ART is linked to low percentages of vRNA+ TFH in PLWH on ART. Duration of ART may be a critical determinant of the cells that harbor the active reservoir and should be considered in the context of cure studies.

TFH in B cell follicles of ART-naïve PLWH were uniquely susceptible to HIV expression. No such predisposition was observed in TFH in B cell follicles of those on prolonged ART or in extrafollicular regions of either group. Abundant FDC-bound viral particles were observed only in lymph nodes from untreated PLWH, whereas no such particles were observed in individuals on prolonged ART. A seminal study previously demonstrated that FDC-bound HIV particles are potently infectious to CD4+ T cells [6]. It is unknown what the rate of decay of the replication competent FDC-bound HIV reservoir is *in vivo*, although some have hypothesized that it could persist for several years [61]. Decay of the FDC-bound reservoir could potentially explain the progressive decrease in CA vRNA in TFH reported by Banga *et al.* during ART. Lack of FDC-bound viral particles in PLWH in our study suggests that FDC-mediated HIV transmission did not play a significant role in vRNA expression in those individuals. Experiments in which viral particles are displaced from FDC could provide important insight into the hypothesized relationship between FDC-bound virus particles and HIV expression in TFH and, if this relationship is established, could be used to potentially accelerate decay of the active reservoir in TFH in PLWH on ART.

In both groups of PLWH, a small fraction of vRNA+ cells in extrafollicular regions had a TFH phenotype. One report hypothesized that pre-TFH become infected in extrafollicular regions, and then migrate into B cell follicles [62]. Our experiments, however, demonstrated that both GCB and HIV expression in TFH are associated with reduced cell surface CXCR5 expression, which is important for access and retention in B cell follicles [23]. Furthermore, HIV infection impaired chemotaxis to CXCL13. Taken together, these findings suggest that HIV-expressing pre-TFH are unlikely to migrate into B cell follicles, whereas both GCB and HIV likely facilitate exit of vRNA+ TFH from follicles into extrafollicular regions of SLT.

There are several limitations of this study. Experiments utilizing tonsil cells from children infected with HIV *ex vivo* do not necessarily replicate all aspects of SLT in adults LWH. Routine tonsillectomies are often performed because of recurrent inflammation or infection, factors known to alter TFH responses [63–64]. Furthermore, there are significant differences in TFH responses among children, young adults, and older adults [65–68]. Experiments utilizing SLT cells from PLWH both on and off ART could be important in solidifying the findings that lymphoid B cells promote HIV replication. While most vRNA+ cells in untreated PLWH are likely recently infected [10], similar to the tonsil model, latently infected cells are thought to be the precursors of most vRNA+ cells in PLWH on ART. Increasing evidence suggests, however, that chronic low level transcription of incomplete transcripts is ongoing in many latently HIV-infected cells during ART [69]. It is plausible that factors, such as B cells, that promote HIV expression in recently infected cells may also induce a transcriptional shift in latently infected cells to promote virus expression. Another limitation is that not all vRNA+ cells harbor replication competent virus. Unfortunately, assays to identify intact virus either at the transcript or genome level *in situ* are not currently available. Regardless, it is reasonable to expect that similar mechanisms are involved in induction of vRNA expression for both intact and defective provirus, and both are hypothesized to contribute to HIV immunopathogenesis. It should be noted that *in situ* assays for vRNA combined with vDNA to assess the role of B cell adjacency in HIV expression were not performed in this study as they were deemed unlikely to be informative due to the high levels of defective integrated vDNA in PLWH in vivo [70–71]. When *in situ* assays for intact vDNA become available, however, they could be used to further solidify the role of B cell adjacency in vRNA expression. Another limitation is that the number of PLWH from whom tissue was analyzed was small, and all were male. It would be important for the findings here to be validated in larger studies including women and men. Finally, there were significant age differences between PLWH on prolonged ART

and those who were untreated in this study; some differences between these groups could be related to age and duration of HIV infection rather than ART. This problem is inherent to all studies comparing ART-treated and ART-naïve PLWH, as individuals with untreated HIV rarely survive as long as those on prolonged ART.

Much attention has been focused upon the role of B cells in generating broadly neutralizing antibodies against HIV. The studies here, however, suggest B cells have a broader role in HIV immunopathogenesis and may play a key role in inducing expression of vRNA and reigniting virus replication when ART is stopped. Other antigen presenting cells such as macrophages and dendritic cells may also play roles in promoting HIV expression *in vivo*, and future studies to investigate their contributions are warranted. Intriguingly, in mice, B cells induce quiescence and memory CD4$^+$ T cell formation, particularly in response to low antigen levels [72], similar to conditions after ART administration. Thus, B cells may promote the formation of the latent viral reservoir after ART initiation. A better understanding of the factors within SLT that contribute to HIV expression and persistence is essential to developing effective HIV eradication strategies.

## Materials and methods

### Ethics Statement

Tonsils were obtained from children at low risk for HIV infection who had undergone routine tonsillectomy. Use of tonsil specimens for these studies was approved by the University of Arizona Institutional Review Board and determined to not constitute human subjects research. Informed consent was not required according to guidelines issued by the Office of Human Research Protections.

Deidentified archived inguinal lymph nodes were obtained from six untreated PLWH without AIDS as previously described [73–74] and from six PLWH on prolonged ART from the University of Hawaii. All PLWH were male, and they were matched based on sex, race, and CD4$^+$ T cell counts. None had an opportunistic infection, malignancy or acute illness at the time of lymph node excision. Use of lymph node specimens for these studies was approved by the University of Arizona Institutional Review Board and determined exempt. Informed consent was not required according to the guidelines issued by the Office of Human Research Protections.

### HIV reporter virus

293T cells grown in DMEM (Gibco) supplemented with 10% fetal bovine serum (FBS), 1X nonessential amino acids (Gibco), 1X Glutamax (Gibco), and 1X Primocin (Invivogen) were transfected with X4-tropic pNLENG1-IRES [26] (X4-HIV) or R5 tropic pNLYUV3-GFP [75] (R5-HIV) virus plasmid using Lipofectamine 2000 (ThermoFisher). Supernatant was collected 2 days post transfection, centrifuged at 800g, and stored in 500 µl aliquots at -80°C prior to use.

### Tonsil cell infections

500µl of virus supernatant or DMEM (Gibco) for mock infections were added to a maximum of 5x10$^6$ cells and spinoculated for 2 hours at 1200g at room temperature. Cells were washed with PBS to removed unbound virus and suspended in R-15 media consisting of RPMI (Gibco) supplemented with 15% heat inactivated FBS, 1X nonessential amino acids (Gibco), 1X Glutamax (Gibco), 1X Primocin (Invivogen), and 5µM saquinavir (Sigma) at a density of 2x10$^6$ cells/ml.

### Tonsil TFH and nonTFH cell isolation

Tonsils were cut into 2–5mm pieces with scalpels in 12ml flow buffer (2% FBS, 2mM EDTA, in PBS). Supernatant was passed through a 100 µm cell strainer and centrifuged at 400g for 10 minutes. Tonsil CD4$^+$ T cells were enriched from disaggregated tonsil tissue using negative selection (STEMCELL Technologies). CD4$^+$ T cells were labeled in 100µl flow buffer with 10µl CD3-AF700 (clone SP34–2, BD), 5µl CD8-BV510 (clone RPA-T8, BioLegend), 5µl CXCR5-PE (clone MU5UBEE, eBioscience), 8µl PD-1-BV785 or BV421 (clone EH12.2H7, Biolegend) for 20 minutes, washed, and

resuspended in flow buffer with 10µl 7-AAD (Tonbo) and sorted using a FACSAriaIII cell sorter equipped with an 85 µm nozzle, or with a Miltenyi Tyto using high speed cartridges. TFH were defined as CD3$^+$CD8$^-$CXCR5$^{hi}$PD-1$^{hi}$7-AAD$^-$ and nonTFH defined as CD3$^+$CD8$^-$CXCR5$^{-/mid}$PD-1$^{-/mid}$7-AAD$^-$ except when 10µl ICOS-BV421 (clone C398.4A, Biolegend) was used in place of CXCR5 to sort TFH where indicated.

## Tonsil B cell isolation

Cell pellets from disaggregated tonsils were suspended in 100µl flow buffer and labeled with 5µl CD3-AF700 (clone SP34–1, BD), 20µl CD19-PECy7 (clone SJ25C1, Tonbo), 10µl IgD-FITC (clone IA6–2, BD), 10µl CD38-APC (clone HIT2, Tonbo) or CD38-BV421 or BV510 (clone HIT2, Biolegend) for 20 minutes, washed, and resuspended in flow buffer with 10µl 7-AAD (Tonbo). B cells were sorted on a FACSAriaIII cell sorter equipped with an 85 µm nozzle, or with a Miltenyi Tyto using high speed cartridges. GCB (CD3$^-$CD19$^+$IgD$^-$CD38$^{mid}$7-AAD$^-$), IgD$^+$CD38$^-$ naïve B cells (CD3$^-$CD19$^+$IgD$^+$CD38$^-$7-AAD$^-$), IgD$^+$CD38$^+$ pre-GCB (CD3$^-$CD19$^+$IgD$^+$CD38$^+$7-AAD$^-$), and IgD$^-$CD38$^-$ memory B (CD3$^-$CD19$^+$IgD$^-$CD38$^-$7-AAD$^-$) populations were isolated where indicated.

## FDC isolation

FDC were isolated from tonsils as described previously [76] with some alterations. Tonsils were mechanically dissociated in RPMI and tissue pieces were digested with one Wünsch unit of Liberase TM (Roche) and 100 U Benzonase (Sigma) for 60 minutes in a 37°C water bath and occasional mixing. Tissue was passed through 25ml and 10ml serological pipettes and an additional Wünsch unit of Liberase TM was added for an additional digestion for 45 minutes. Digested tissue was passed through a 100 µm cell strainer and the enzymatic digestion was quenched by the addition of FBS. Tissue-free cell supernatant was centrifuged at 400g for 10 minutes and resuspended in R-15 media. Cells were layered on top of a 43% percoll solution and centrifuged for 20 minutes at 1000g. Cells were collected from above the percoll solution and washed in PBS. Cells were resuspended in 1ml flow buffer and labeled with 10µl CD35-biotin (Miltenyi) and 10µg LFA-1 blocking antibody (ThermoFisher) and incubated overnight at 4°C. Cells were washed in flow buffer and subjected to CD45 depletion by positive selection (STEMCELL Technologies). Unbound cells were labeled in 100µl flow buffer with 10µl CD45-PE (clone HI30, Tonbo), 5µl Streptavidin-BV421 (BD) and incubated for 20 minutes, washed, and resuspended in flow buffer with 10µl 7-AAD. FDC (CD45$^-$CD35$^+$7-AAD$^-$) were isolated on a cell sorter and cultured with spinoculated TFH where indicated.

## Cell labeling for cocultures

Uninfected cells were labeled with 1µl CellTrace Blue (ThermoFisher) or 1µl Violet Proliferation Dye (ThermoFisher) in 1ml PBS for 10 minutes at 37°C. Labeling was quenched by addition of R-15. Labeled cells were washed in PBS and suspended in R-15 at 2x10$^6$ cells/ml.

## Cell culture

Spinoculated TFH and nonTFH were cultured 1:1, except where indicated, with autologous uninfected, CellTrace Blue, or violet proliferation dye-labeled B cell subsets, TFH, or nonTFH, at a final concentration of 2x10$^6$ cells/ml for three days in R-15. In some experiments 5µg/ml CXCL13 blocking antibody (clone AF801, R&D systems), 10µg/ml anti-IL-2 (clone 5334, R&D systems), 10µg/ml anti-GITRL (clone 109114, R&D systems), 5µg/ml 4–1BB-Fc (R&D systems), 10µg/ml anti-MHC-II (clone IVA12, Raybiotech), 10µg/ml sICOS (Biolegend), 10µg/ml sCD40L (Biolegend), 10µg/ml anti-CD40 (clone 82102, R&D systems), 100nM AZD5582 (SelleckChem), 5µM NIK-SMI1 (Sigma), 0.1-1µM tofocitinib (SelleckChem), 0.1-1µM ruxolitinib (SelleckChem), or DMSO as an appropriate vehicle control when necessary were added. Where indicated spinoculated cells were physically separated from autologous uninfected TFH or GCB using transwell inserts with 0.4 µm pores (Corning).

PLOS Pathogens

## CD19⁺ cell depletion and culture

Disaggregated tonsil cells were suspended in flow buffer and depleted of CD19$^+$ cells using an anti-biotin microbeads, ultrapure kit (Miltenyi) and CD19-biotin antibody (clone SJ25C1, Tonbo). Cells were counted in the effluent and eluent to determine the CD19$^-$ to CD19$^+$ ratio. $5 \times 10^6$ CD19$^-$ cells were spinoculated with X4-HIV or R5-HIV. Uninfected CD19$^-$ and CD19$^+$ cells were labeled with CellTrace Blue and cultured with autologous spinoculated cells at the donor-specific ratio in a final concentration of $2 \times 10^6$ cells/ml in R-15. Cells were cultured three days and GFP expression was measured by flow cytometry.

## Flow cytometry

Cells were washed in PBS and resuspended in 50µl PBS and 0.5µl Ghost Dye Red 780 (Tonbo). After 2 minutes, antibodies in 50µl PBS were added and incubated for 20 minutes in the dark at room temperature. Cells were washed in PBS and resuspended in 2% paraformaldehyde in PBS. CountBright absolute counting beads (Molecular Probes) were used for cell count determination. For phenotyping, the following antibodies were used: 2µl CD3-AF700 (Clone SP34–2, BD), 2µl CD8-BV510 (Clone RPA-T8, BioLegend), 2µl PD-1-BV421 or BV785 (Clone EH12.1H7, BioLegend), 5µl ICOS-BV421 (clone C398.4A, Biolegend), 2µl CXCR5-PE (Clone MU5UBEE, eBioscience), 2µl CD19-PECy7 (clone SJ25C1, Tonbo). Cells were analyzed on a Fortessa flow cytometer, and data analyzed using FloJo v10.

## Quantitative polymerase chain reaction (QPCR) for HIV DNA quantification

GCB, spinoculated TFH, and spinoculated TFH with GCB were cultured for 18 hours. GCB cultured alone were added to TFH cultured alone immediately prior to DNA isolation to maintain equivalent cell numbers. DNA was isolated using a Puregene kit (Qiagen). Cellular DNA was quantified by QPCR for cell equivalents of DNA per volume as described previously [77]. Total and integrated DNA were quantified as described previously [78] using equal quantities of cellular DNA.

## Gene expression analysis

TFH were isolated from 6 tonsils, mock spinoculated, or spinoculated with R5-tropic HIV GFP reporter virus, and cultured with uninfected, violet proliferation dye (ThermoFisher) labeled TFH or GCB for 3 days in R-15 with 5µM saquinavir. Identical conditions were prepared for nonTFH spinoculated with R5-tropic HIV GFP reporter virus, and cultured with either uninfected labeled nonTFH or nonGCB. After 3 days, cells were washed and labeled with 10µl 7-AAD in flow buffer. Live, unlabeled TFH or nonTFH cells were isolated on a cell sorter. In a separate experiment, TFH were spinoculated with X4-tropic HIV GFP reporter virus, and cultured alone or with GCB in R-15 with saquinavir. After 3 days, cells were labeled with 10µl CD3, 10µl CD19 in 100µl flow buffer for 20 minutes, washed, and resuspended in flow buffer containing 10µl 7-AAD and live TFH were isolated on a cell sorter. Percent GFP and GFP MFI were also measured from a minimum of $1 \times 10^6$ events during the sort. RNA was isolated from sorted TFH using Picopure RNA isolation kit (ThermoFisher). RNA concentrations were determined using Qubit and quality was determined using a 5200 Fragment Analyzer. 83–100 ng RNA were input for gene expression analysis using the Host Response Panel (Nanostring) and an nCounter (Nanostring). Analysis was performed using nSolver 4.0. Eight housekeeping genes in the panel were used for normalization and selected based on the average counts and percent CV. Of the 785 genes evaluated, at least 520 were detected above background.

## CXCL13 ELISA

TFH and GCB cell cultures were prepared as described above. After 3 days in culture, cell supernatant was collected and stored at -80°C. Cells were stained, counted, and analyzed as indicated above. Cell counts were used to determine the

quantity of live spinoculated TFH. CXCL13 was quantified from cell culture supernatant (Human BLC ELISA, RayBiotech) and CXCL13 concentrations were normalized to TFH cell counts.

**CXCL13 chemotaxis assays**

TFH were isolated and spinoculated as indicated above. Half of the spinoculated cells were immediately put into culture for 3 days in R-15 with saquinavir. The remaining cells were placed in 5μm transwells in 100μl and transwell inserts were placed into 24 well plates containing 600μl RPMI or RPMI+3μg/ml CXCL13. After 4 hours, inserts were removed and cells from inserts and wells were washed and placed in separate wells with fresh R-15. After 3 days, cells were stained as described above and percentages of GFP$^+$ cells and absolute cell counts were determined. On day 3, chemotaxis assays were performed on cells placed immediately into culture post spinoculation. Cells from inserts and bottom of wells were stained directly after the 4 hour incubation. Absolute counting beads and flow cytometry were used to determine percentages of GFP$^+$ and GFP$^-$ cells in each compartment. Chemotaxis indices were calculated by subtracting the mean percentages of GFP$^+$ or GFP$^-$ cells that migrated in the absence of chemokine from the percentages of GFP$^+$ or GFP$^-$ cells, respectively, that migrated in the presence of 3μg/ml CXCL13.

***In situ* hybridization and immunostaining of paraffin embedded lymph nodes**

Formalin fixed, paraffin embedded lymph node sections were stained fluorescently using a combination of multiplex *in situ* hybridization (RNAscopeTM, ACD) and immunofluorescent staining. Briefly, 6μm thick sections of paraffin embedded tissues were mounted onto slides (2 – 3 sections per slide, 60 μm apart) and air dried overnight. Slides were deparaffinized using 2 changes of Xylene (5 min. each) and 2 changes of 100% ethanol and air dried. All samples were treated for 15 min. in 1% $H_2O_2$, washed in deionized water, subjected to antigen retrieval with 1X antigen retrieval buffer (ACD) for 30 min. in a > 90°C water bath, and treated with protease (ACD Protease Plus) for 30 min. at 40°C. Probes (ACD) were added and incubated for 2 hours followed by detection steps using the ACD multiplex kit and labeled with fluorescent tyramide labeling reagent (Invitrogen). This was followed by overnight labeling with primary antibody and detected using fluorescently labeled secondary antibodies (Invitrogen). In some cases directly labeled CD20 (ZenonTM labeling reagent, Invitrogen) was used as a final step and all slides were counterstained with DAPI and mounted with SlowFadeTM Diamond (Invitrogen). Sections were imaged on a Leica Aperio Versa 8 slide scanning system outfitted with appropriate filters for detection of AF488, AF555, AF594, AF647 and DAPI. Z-stacks were used where indicated. The slides were scanned using the Aperio Versa Application Software v1.0.2 with an ANDOR Zyla sCMOS camera at 40X (NA 0.85) and room temperature. Digital files were analyzed using Image Scope (Leica, v.12.4.0.5043).

**Data analyses of stained tissue sections**

Follicle and GC sizes were determined from sections stained for CD20 (1:50, Rab mAb, ab78237, Abcam) and FDC (1:50, clone CNA.42, 14-9968-37, Invitrogen). Total tissue area, follicular area defined by CD20 staining, and germinal center area defined by FDC staining were determined using Image Scope. vRNA$^+$ cell frequency, location, and CD20 adjacency, and CD20 frequency quantifications were performed using visual inspection of sections stained with HIV-1 probe (V-HIV1-Clade B, 416111, ACD) and CD20 antibody. Frequencies of CD20 cells in extrafollicular regions were determined after counting all DAPI$^+$ cells and CD20$^+$DAPI$^+$ cells within 10 areas of approximately 50 cells randomly selected in up to three nonadjacent sections. The probability of a vRNA$^+$ cell being adjacent to at least one B cell in extrafollicular regions assumed random distribution of B cells within the extrafollicular regions and one vRNA$^+$ cell adjacent to six other cells, as previously described [66]. The probability was calculated as {100% - (%CD20 negative cells)$^6$}. TFH phenotyping of vRNA$^+$ cells was done on a subset of sections stained with HIV-1 probe, BCL6 probe (Hs-BCL6-C2, 407251-C2, ACD), PD-1 antibody (1:50, Rab mAb, ab237728, Abcam) and CD20 antibody. Each vRNA$^+$ cells was assessed for expression of markers

for TFH defined as *BCL6*[+]PD-1[+]. Analysis of TFH and CD4[+]T cell frequency was performed on sections stained with CD4 probe (1:50, Hs-CD4-C3, 605601-C3, ACD), BCL6 probe, PD-1 antibody, and CD3 antibody (MS mAb, ab17143, Abcam). Frequencies of TFH and CD3[+]*CD4*[+]T cells were determined similarly to CD20 frequencies as above. Follicular areas were determined using an adjacent section stained for CD20. TFH were defined as CD3[+]*CD4*[+]PD-1[+]*BCL6*[+] and CD4[+]T cells were defined as CD3[+]*CD4*[+]. Data from multiple sections were combined for each subject.

## Statistical analysis

Comparisons of tonsil cultures were performed using nonparametric Friedman tests and Dunn's multiple comparison tests or Wilcoxon tests as determined by Graphpad Prism v10. Fold changes and p values for gene expression analyses were calculated using nSolver 4.0 as per Nanostring's recommendations. Adjusted p value calculations were determined using the Benjamini-Hochberg method of estimating false discovery rates and accompanying plots were prepared using Graph-Pad Prism v10.2.2. Directed global significance scores using adjusted p values were determined using nSolver advanced analysis using fast settings as recommended by Nanostring. Comparisons of PLWH on ART and PLWH without ART were performed by nonparametric Mann-Whitney tests, or in direct comparisons of paired data, a paired Wilcoxon test was performed. Significance is denoted in each figure by asterisks, as $*p < 0.05$; $**p \leq 0.01$; $***p \leq 0.001$; $****p \leq 0.0001$.

## Supporting information

**S1 Fig. Supporting data for Fig 1.** (A) Dose response curves for GCB-induced HIV expression. $2 \times 10^5$ TFH were spinoculated with X4-HIV and cultured with autologous uninfected CellTrace Blue labeled GCB at the ratios indicated. Percent GFP[+]TFH and GFP MFI of GFP[+]TFH were determined by flow cytometry. These data were used to determine fold differences as reported in Fig 1E. (B) Cell counts of uninfected, CellTrace labeled TFH and GCB after 3 days in culture with HIV spinoculated TFH as reported in Fig 1H.
(PDF)

**S2 Fig. Gene expression analyses of cultures of TFH with either TFH or GCB and examination of several mechanisms by which GCB could potentially upregulate HIV expression in TFH.** (A) Upregulation of GFP expression in TFH (CD3[+]CD8[-]CD19[-]CXCR5[hi]PD-1[hi]) by GCB (CD19[+]CD3[-]CD38[mid]IgD[-]) for experiments reported in Figs 2B and S2D. (B-D) Volcano plots depicting differentially expressed genes from (B) R5-HIV GFP reporter virus (R5-HIV) spinoculated TFH cultured with uninfected, dye labeled TFH (R5-HIV TFH + TFH) compared to mock spinoculated TFH cultured with dye-labeled TFH (Mock TFH + TFH), (C) R5-HIV spinoculated TFH cultured with dye-labeled GCB (R5-HIV TFH + GCB) compared to mock spinoculated TFH cultured with dye-labeled GCB (Mock TFH + GCB), and (D) X4-HIV spinoculated TFH cultured with GCB (X4-HIV TFH + GCB) compared to X4-HIV spinoculated TFH (X4-HIV TFH). Vertical lines represent 1.5 fold change and horizontal lines represent an adjusted p value of 0.05, as determined using the Benjamini Hochberg method. (E) Heatmap of the 126 genes altered significantly in at least one of the three cultures reported in Figs 2A, 2B and S2D. (F) The number of TFH genes significantly altered by GCB that were shared or distinct in X4-HIV (grey), R5-HIV (green), or Mock (pink) cultures as reported in (D), Fig 2B and 2A, respectively. (G) TFH were sorted from CD4 enriched tonsil cells in which CD3 antibody was either included (CD3[+]CD8[-]CD19[-]CXCR5[hi]PD-1[hi]) or omitted (CD8[-]CD19[-]CXCR5[hi]PD-1[hi]) in autologous preparations. TFH were spinoculated with X4-HIV and cultured with uninfected labeled TFH or GCB for 3 days in R-15 and 5µM saquinavir. GCB-mediated fold differences in percentages of GFP[+]TFH (top) and GFP MFI of GFP[+]TFH (bottom) were determined (n = 6). (H-L) TFH isolated from tonsils were spinoculated with X4-HIV and cultured with uninfected, CellTrace Blue labeled TFH or GCB at a ratio of 1:1. Cells were cultured in the presence or absence of (H) soluble ICOS, sCD40L, or blocking antibody to CD40 (n = 6), (I) with or without blocking antibody to ICAM-1 (n = 4), (J) in the presence or absence of neutralizing antibody to IL-2, GITRL, or 4–1BB-Fc (n = 3), (K) DMSO, 0.1µM of the noncanonical NFκB activator AZD5582, 5µM of the non-canonical NFκB inhibitor NIK-SMI1, or both

AZD5582 and NIK-SMI1 (n = 6), or (L) DMSO, and either 0.1μM or 1μM of JAK inhibitors tofacitinib (Tof) or ruxolitinib (Rux) (n = 2). After 3 days, percent CellTrace Blue$^-$GFP$^+$TFH and GFP MFI of CellTrace Blue$^-$GFP$^+$TFH were determined by flow cytometry and reported as GCB-mediated fold differences. (M) GCB-mediated fold differences of experiments as reported in Fig 2D. Horizontal bars indicate medians (G-L). Statistical analyses of (A,G,H,K,M) were determined using Wilcoxon matched paired tests as determined by Graphpad Prism v10 and significance indicated: ns, not significant; *p < 0.05.
(PDF)

**S3 Fig. Representative gating strategy for isolation of TFH by FACS in which ICOS antibody was included and CXCR5 antibody omitted as used in Fig 3D.**
(PDF)

**S4 Fig. SLT B cells upregulate HIV expression in TFH and nonTFH and alter gene expression in nonTFH.** (A) 4x10$^6$ nonTFH spinoculated with R5-HIV GFP reporter virus were cultured with 4x10$^6$ uninfected, violet proliferation dye labeled nonTFH (R5-HIV nonTFH + nonTFH) or nonGCB (R5-HIV nonTFH + nonGCB) for 3 days in R-15 with 5μM saquinavir. Live, violet proliferation dye$^-$ nonTFH were isolated by FACS and total RNA was purified. Gene expression analysis was performed using Nanostring's host response panel. Volcano plot depicts differentially expressed genes in R5-HIV nonTFH + nonGCB when compared to R5-HIV nonTFH + nonTFH. Vertical lines represent 1.5 fold change and horizontal lines represent an adjusted p value of 0.05, as determined using the Benjamini Hochberg method. (n = 6) (B) Directed global significance scores were determined for the 30 pathways in which ≥15 genes from the host response panel were assigned using nSolver 4.0 advanced analysis. (C) The number of genes significantly altered by B cells that were shared or distinct in R5-HIV nonTFH (blue), Mock TFH (pink), R5-HIV TFH (green), and X4-HIV TFH (grey) cultures as reported in (A), Figs 2A, 2B, and S2D, respectively.
(PDF)

**S5 Fig. Representative imaging of vRNA$^+$TFH and TFH cells, and frequencies of T cell subsets in lymph nodes from PLWH.** (A) Representative images of inguinal lymph node sections illustrating vRNA$^+$TFH defined as DAPI$^+$vRNA$^+$B-CL6$^+$PD-1$^+$ inside (top) and outside (bottom) B cell follicles, as determined by CD20 staining (not shown). All *in situ* targets were imaged using z-stacks and shown as compressed images. (B) Percentages of vRNA$^+$DAPI$^+$ cells expressing *BCL6*, PD-1, and both in PLWH. (C) Representative single color images of the same TFH stained by immunostaining for CD3 (top left), PD-1 (bottom left), and *in situ* hybridization for *CD4* (top right), and *BCL6* (bottom right). DAPI (blue) was used to identify nuclei. Arrows indicate the same TFH cell. (D) Frequencies of CD3$^+$CD4$^+$T cells were determined in follicles (F) and extrafollicular (EF) regions in PLWH. Horizontal bars indicate medians (D). Mann-Whitney tests were performed for unpaired comparisons between PLWH on ART (+ART) and PLWH not on ART (No ART), and Wilcoxon tests were used for paired comparisons as determined by Graphpad Prism v10 and significance indicated: ns, not significant; *p < 0.05.
(PDF)

**S1 Table. Numbers of vRNA$^+$ cells evaluated from each donor.**
(PDF)

**S1 Data. Raw and analyzed gene expression data used in Figs 2 and S2.**
(ZIP)

# Acknowledgments

Cell sorting was performed at the University of Arizona Cancer Center Flow Cytometry Shared Resource supported by the National Cancer Institute (P30 CA023074).

## Author contributions

**Conceptualization:** Matthew T. Ollerton, Joy M Folkvord, Elizabeth Connick.

**Data curation:** Joy M Folkvord.

**Formal analysis:** Matthew T. Ollerton, Joy M Folkvord, Veronica Bush.

**Investigation:** Matthew T. Ollerton, Joy M Folkvord.

**Methodology:** Matthew T. Ollerton, Joy M Folkvord.

**Project administration:** Elizabeth Connick.

**Resources:** David A. Parry, Amie L. Meditz, Martin D McCarter, Fred Yost, Cecilia M Shikuma.

**Supervision:** Elizabeth Connick.

**Validation:** Matthew T. Ollerton, Joy M Folkvord.

**Visualization:** Matthew T. Ollerton, Joy M Folkvord, Elizabeth Connick.

**Writing – original draft:** Matthew T. Ollerton, Elizabeth Connick.

**Writing – review & editing:** Matthew T. Ollerton, Joy M Folkvord, Veronica Bush, David A. Parry, Amie L. Meditz, Martin D McCarter, Fred Yost, Cecilia M Shikuma, Elizabeth Connick.

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
