## [Decision Letter · Decision Letter 0]

21 Aug 2025

Lymphoid B cells upregulate HIV-1 ex vivo and are linked to its expression in vivo

PLOS Pathogens

Dear Dr. Ollerton,

Thank you for submitting your manuscript to PLOS Pathogens. After careful consideration, we feel that it has merit but does not fully meet PLOS Pathogens's publication criteria as it currently stands. Therefore, we invite you to submit a revised version of the manuscript that addresses the points raised during the review process.

Please submit your revised manuscript within 60 days Oct 20 2025 11:59PM. If you will need more time than this to complete your revisions, please reply to this message or contact the journal office at plospathogens@plos.org. Please include the following items when submitting your revised manuscript:

We look forward to receiving your revised manuscript.

Kind regards,

Richard A. Koup, M.D.

Section Editor

PLOS Pathogens

Richard Koup

Section Editor

PLOS Pathogens

Editor-in-Chief

PLOS Pathogens

orcid.org/0000-0003-2946-9497

Editor-in-Chief

PLOS Pathogens

orcid.org/0000-0002-7699-2064

**Journal Requirements:**

At this stage, the following Authors/Authors require contributions: Matthew Ollerton, Joy M Folkvord, Veronica Bush, David A. Parry, Amie L. Meditz, Martin D McCarter, Fred Yost, Cecilia M Shikuma, and Elizabeth Connick. Please ensure that the full contributions of each author are acknowledged in the "Add/Edit/Remove Authors" section of our submission form.

4) We have noticed that you have cited Table  1 in the manuscript file but there is no corresponding table in the manuscript.  Please amend your manuscript to include this table noting that tables should not be uploaded as individual files.

2) If any authors received a salary from any of your funders, please state which authors and which funders..

6) Kindly revise your competing statement to align with the journal's style guidelines: 'The authors declare that there are no competing interests.'

**Reviewers' Comments:**

Reviewer's Responses to Questions

**Part I - Summary**

Reviewer #1: In this manuscript the authors are combining ex vivo and in-vivo (in situ) data to show the involvement of B cells in HIV genome replication. Using tonsils and flow cytometry approach, the authors are showing that GC B cells induced HIV genome expression in TFH in a concentration, contact and MHC-II dependent manner. Then using LN biopsies isolated from people with HIV (on ART and not treated) and by performing in situ staining for vRNA and different cell markers, the authors shown that vRNA+ cells numbers were elevated in B cell follicles, but TFH were a minority of the infected cells in both groups. They also shown that most vRNA+ were preferentially adjacent to extrafollicular B cells.

Reviewer #2: The manuscript “Lymphoid B cells upregulate HIV-1 ex vivo and 1 are linked to its expression in vivo” by Ollerton et all examines the impact of viral RNA expression in HIV-infected TFH by B cells. The authors use an in vitro model of HIV infection using cells derived from adolescent tonsils and ex vivo analysis of lymph nodes from adult PWH on or off ART. The authors characterize HIV expression in the presence or absence of GCB and use transcriptomics, pathway blockade and LRAs to probe the mechanisms. The authors conclude GCB increase the expression of HIV in TFH in a contact and MCH II dependent manner. The manuscript is well written, and the figures are clear.

**Part II – Major Issues: Key Experiments Required for Acceptance**

Reviewer #1: This manuscript is really well written, designed and is investigating an interesting parameter often forgotten in the field which is the effect of B cells on infected cells.

I had great pleasure reading this manuscript, but I would like to raise a few concerns and make suggestions that hopefully can improve this interesting work.

1- Tonsils are a great tissue source however, if tonsillectomy is performed it is usually because of recurrent inflammation/infections. The authors are not raising this limitation in their discussion, and I think it would be important to mention that the levels of activation (in B and T cells) in those samples are likely not normal and can potentially affect results and increase cell death ex vivo and secretion of cytokines and other important factors.

2- Sorry if I missed it in the method but, it was difficult for me to understand if ex vivo the isolated GCB cells and Tfh put together in culture were from the same donor or donors were mixed – as this can highly affect the response/activation of the cells.

3- One important data missing in the ex vivo culture is the viability of the B cells- B cells aren’t the easiest cells to culture and are highly sensitive so can die rapidly which would explain the results obtained in culture and would also be dose (or ratio) dependent. The authors shown the viability of TFH but not of the GCB cells.

4- I was a little confused by the number of samples analyzed and the “n” reported in graphs – for example in the text the authors are talking about 6 samples being analyzed and flow data have often n= 7 or 1. Can the authors clarify this? Is it the flow analysis being repeated on same samples? If so can it be indicated on graphs if same sample is represented in different runs.

5- Regarding the in situ work, I have great concerns regarding the pattern of the representative pictures presented in figure S5 and Figure 6. First, the in situ CD4 is a single indicating a DNA signal (as all RNAprobe will detect complementary DNA but pattern will be a single dot) and not an expression of CD4 in that particular cell- I hope the authors took this into consideration while quantifying their cells of interest. The vRNA signal in figure S5 is quite unusual (figure 6 is more what is expected). Finally using in situ for BCL6 imply a need for membrane marker to determine an accurate segmentation of the cell and so being able to identify which dot belongs to which cells- I might have missed it, but I didn’t see anything regarding cell segmentation for accurate phenotyping in the method.

6- The quantification of B cells in all patients, within and out of BCF needs to be added to this manuscript to see if in those patients the repartition of B cells and their pattern is similar before performing any comparison between group and tissue areas.

7- There is a great difference in cell density in BCF (especially GC) and T cell zone or medullary cords and this can change drastically with inflammation and treatment. Did the authors performed nuclei count and can express their data per million nuclei instead of mm2 of tissue to see if the conclusion and patterns stay similar.

8- Running a duplex (vRNA+vDNA) would have been more accurate and powerful to compare treated and not treated patients with the hypothesis of B cells increasing genome expression.

Reviewer #2: 1. In Figure 2 the authors perform transcriptomics analysis on TFH cells in vitro infected with X4 or R5 tropic HIV in the presence or absence of GCB. Although the authors show the data in Figure 2 and report the gene expression measurements in xcel files, the authors seem to be leaving it up to the reader to review and interpret their findings. There is really no discussion of the findings in the manuscript other than to note a few genes in lines 128-131. The authors should include a more detailed discussion on the gene signatures they are showing in Figure 2 and more discussion on the significance of these findings in their conclusions. Do the authors observe any correlations between the magnitude of the change in gene expression and the magnitude of GFP? Or magnitude of the HIV RNA measured in the RNAseq? The authors try to block some of the pathways observed in the transcriptomics data but do not observe the expected impact on GCB mediated induction of HIV expression in TFH. What do the authors propose as an explanation for these results?

2. For the in vitro experiments the authors use primary TFC, FDC and GCB derived from tonsils harvested from pediatric donors. Given the difference in immunological profile of Tfh cells between adolescents and adults, how generalizable are the observations made to adults in these experiments? Or are these observations specific to pediatric HIV infection? For example, it has been shown that Tfh cells exhibit age dependent altered tissue localization and transcriptional programming (Immun Ageing. 2014;11:12, Nat Immunol. 2023;24(7):1124–1137, Elife. 2021;10:e70554). Would the authors expect a similar result using adult derived tonsil MNCs? Although the authors use adult LN from men living with HIV on and off ART in the final figure to validate their findings, there is no discussion of how age related changes in GCB and TFH from their in vitro experiments would impact there conclusions to adult PWH and this should be noted and discussed.

**Part III – Minor Issues: Editorial and Data Presentation Modifications**

Reviewer #1: (No Response)

Reviewer #2: (No Response)

PLOS authors have the option to publish the peer review history of their article (what does this mean? ). If published, this will include your full peer review and any attached files.

**Do you want your identity to be public for this peer review?** For information about this choice, including consent withdrawal, please see our Privacy Policy .

Reviewer #1: No

Reviewer #2: No

**Figure resubmission:**

**Reproducibility:**



---

## [Decision Letter · Decision Letter 1]

23 Oct 2025

Dear Dr Ollerton,

We are pleased to inform you that your manuscript 'Lymphoid B cells upregulate HIV-1 ex vivo and are linked to its expression in vivo' has been provisionally accepted for publication in PLOS Pathogens.

Best regards,

Susan R. Ross, PhD

Section Editor

PLOS Pathogens

Susan Ross

Section Editor

PLOS Pathogens

Sumita Bhaduri-McIntosh

Editor-in-Chief

PLOS Pathogens

orcid.org/0000-0003-2946-9497

Michael Malim

Editor-in-Chief

PLOS Pathogens

orcid.org/0000-0002-7699-2064

Reviewer Comments (if any, and for reference):

Reviewer's Responses to Questions

**Part I - Summary**

Reviewer #1: (No Response)

Reviewer #2: The revised manuscript “Lymphoid B cells upregulate HIV-1 ex vivo and 1 are linked to its expression in vivo” by Ollerton et al addressed key concerns of all reviewers and strengthened their manuscript. In particular, including discussion of the study limitations and opportunities for further investigation are well received. Excellent work.

**Part II – Major Issues: Key Experiments Required for Acceptance**

Reviewer #1: The authors have answered all reveiwers concerns and have added to the manuscript all needed information for a better understanding of the study and data interpretation.

Reviewer #2: No further experiments required.

**Part III – Minor Issues: Editorial and Data Presentation Modifications**

Reviewer #1: NA

Reviewer #2: (No Response)

PLOS authors have the option to publish the peer review history of their article (what does this mean? ). If published, this will include your full peer review and any attached files.

**Do you want your identity to be public for this peer review?** For information about this choice, including consent withdrawal, please see our Privacy Policy .

Reviewer #1: **Yes: ** Claire Deleage

Reviewer #2: **Yes: ** Deanna Kulpa

---

## [Editor Report · Acceptance letter]

Dear Dr Ollerton,

We are delighted to inform you that your manuscript, "Lymphoid B cells upregulate HIV-1 ex vivo and are linked to its expression in vivo," has been formally accepted for publication in PLOS Pathogens.

Best regards,

Sumita Bhaduri-McIntosh

Editor-in-Chief

PLOS Pathogens

orcid.org/0000-0003-2946-9497

Michael Malim

Editor-in-Chief

PLOS Pathogens

orcid.org/0000-0002-7699-2064